# SARS-CoV-2 clade dynamics and their associations with hospitalisations during the first two years of the COVID-19 pandemic

Taavi Päll[1]☯, Aare Abroi[2]☯, Radko Avi[1], Heiki Niglas[3], Arina Shablinskaja[1], Merit Pauskar[1], Ene-Ly Jõgeda[1], Hiie Soeorg[1], Eveli Kallas[1], Andrio Lahesaare[4], Kai Truusalu[1], Dagmar Hoidmets[1], Olga Sadikova[3], Kaspar Ratnik[4], Hanna Sepp[3], Liidia Dotsenko[3], Jevgenia Epštein[3], Heleene Suija[3], Katrin Kaarna[5,6], Steven Smit[7], Lili Milani[7], Mait Metspalu[7], Ott Eric Oopkaup[8], Ivar Koppel[8], Erik Jaaniso[9], Ivan Kuzmin[8], Heleri Inno[8], Uku Raudvere[8], Mari-Anne Härma[3], Paul Naaber[1,4], Tuuli Reisberg[7], Hedi Peterson[9], Ulvi Gerst Talas[8], Irja Lutsar[1], Kristi Huik[1]*

1 Department of Microbiology, Faculty of Medicine, Institute of Biomedicine and Translational Medicine, University of Tartu, Tartu, Estonia, 2 Faculty of Science and Technology, Institute of Technology, University of Tartu, Tartu, Estonia, 3 Department of Communicable Diseases, Health Board, Tallinn, Estonia, 4 SYNLAB Eesti OÜ, Tallinn, Estonia, 5 Clinical Research Centre, Faculty of Medicine, Institute of Clinical Medicine, University of Tartu, Tartu, Estonia, 6 Tartu University Hospital, Tartu, Estonia, 7 Institute of Genomics, Faculty of Science and Technology, University of Tartu, Tartu, Estonia, 8 High Performance Computing Centre, Faculty of Science and Technology, Institute of Computer Science, University of Tartu, Tartu, Estonia, 9 Institute of Computer Science, Faculty of Science and Technology, University of Tartu, Tartu, Estonia

☯ These authors contributed equally to this work.
* kristi.huik@ut.ee

**Data Availability Statement:** The raw sequence data of analyzed samples were submitted to the European Nucleotide Archive under Project accessions: Accession Description PRJEB51354

## Abstract

### Background

The COVID-19 pandemic was characterised by rapid waves of disease, carried by the emergence of new and more infectious SARS-CoV-2 virus variants. How the pandemic unfolded in various locations during its first two years has yet to be sufficiently covered. To this end, here we are looking at the circulating SARS-CoV-2 variants, their diversity, and hospitalisation rates in Estonia in the period from March 2000 to March 2022.

### Methods

We sequenced a total of 27,550 SARS-CoV-2 samples in Estonia between March 2020 and March 2022. High-quality sequences were genotyped and assigned to Nextstrain clades and Pango lineages. We used regression analysis to determine the dynamics of lineage diversity and the probability of clade-specific hospitalisation stratified by age and sex.

### Results

We successfully sequenced a total of 25,375 SARS-CoV-2 genomes (or 92%), identifying 19 Nextstrain clades and 199 Pango lineages. In 2020 the most prevalent clades were 20B and 20A. The various subsequent waves of infection were driven by 20I (Alpha), 21J (Delta) and Omicron clades 21K and 21L. Lineage diversity via the Shannon index was at its highest

during the Delta wave. About 3% of sequenced SARS-CoV-2 samples came from hospitalised individuals. Hospitalisation increased markedly with age in the over-forties, and was negligible in the under-forties. Vaccination decreased the odds of hospitalisation in over-forties. The effect of vaccination on hospitalisation rates was strongly dependent upon age but was clade-independent. People who were infected with Omicron clades had a lower hospitalisation likelihood in age groups of forty and over than was the case with pre-Omicron clades regardless of vaccination status.

## Conclusions

COVID-19 disease waves in Estonia were driven by the Alpha, Delta, and Omicron clades. Omicron clades were associated with a substantially lower hospitalisation probability than pre-Omicron clades. The protective effect of vaccination in reducing hospitalisation likelihood was independent of the involved clade.

## Introduction

As was largely the case in other countries, Estonia experienced multiple COVID-19 waves following the emergence of the pandemic. The first wave to be caused by the SARS-CoV-2 virus occurred between February and June 2020, with a peak hospitalisation of around twelve COVID-19 patients per 100,000 people per day in April 2020, and this was accompanied by a national lockdown which lasted for nine weeks. The next waves were driven by the emergence of Alpha, Delta, and Omicron variants of concern (VOC) [1–5], with three peaks of hospitalisation of between fifty and sixty COVID-19 patients per 100,000 people per day in March 2021, November 2021, and February 2022 respectively (https://www.terviseamet.ee/en/coronavirus-dataset). A partial lockdown was established between March and May 2021 in response to overwhelming hospitalisation levels. The compulsory use of face masks was implemented between August 2021 and April 2022, until all restrictions were lifted in June 2022. Vaccination against SARS-CoV-2 has been available for essential healthcare and social services workers and risk groups since 27 December 2020, followed by those over the age of forty in May 2021, and the 12–15 year-old group in June 2021. VOCs have been associated with COVID-19 disease severity and transmissibility. The latter increased between Alpha, Delta, and Omicron [6], but Omicron was shown to cause less severe disease [7]. There were two major peaks of hospitalisation, in Estonia in October-November 2021 and February-March 2022 respectively. Intensive care unit admissions peaked in Estonia in October-November 2021 (https://www.terviseamet.ee/en/coronavirus-dataset), suggesting that SARS-CoV-2 virus variants which were circulating in Estonia during those waves could be associated with more severe disease outcomes. We can report the results here from the SARS-CoV-2 whole genome sequencing study which was carried out in Estonia between March 2020 and March 2022. We undertook this study in order to be able to understand how distinct SARS-CoV-2 clades influenced the severity of COVID-19 and hospitalisation, and which other factors—including immunisation against SARS-CoV-2—served to contribute to this process.

## Methods

### Study design

The study used SARS-CoV-2 positive PCR test sample leftovers for the purpose of genotyping. The samples were collected and stored by the Estonian Health Board, Synlab Eesti OÜ, and

sequencing of SARS-CoV-2 in Estonia PRJEB47583 Whole genome sequencing of SARS-CoV-2 in Estonia PRJEB47944 Whole genome sequencing of SARS-CoV-2 in Estonia PRJEB48700 Whole genome sequencing of SARS-CoV-2 in Estonia PRJEB50534 Whole genome sequencing of SARS-CoV-2 in Estonia PRJEB50537 Whole genome sequencing of SARS-CoV-2 in Estonia PRJEB47326 Whole genome sequencing of SARS-CoV-2 in Estonia PRJEB47434 Whole genome sequencing of SARS-CoV-2 in Estonia PRJEB47436 Whole genome sequencing of SARS-CoV-2 in Estonia PRJEB47570 Whole genome sequencing of SARS-CoV-2 in Estonia PRJEB48138 Whole genome sequencing of SARS-CoV-2 in Estonia PRJEB48892 Whole genome sequencing of SARS-CoV-2 in Estonia PRJEB50275 Whole genome sequencing of SARS-CoV-2 in Estonia PRJEB50429 Whole genome sequencing of SARS-CoV-2 in Estonia PRJEB50485 Whole genome sequencing of SARS-CoV-2 in Estonia PRJEB46782 Whole genome sequencing of SARS-CoV-2 in Estonia PRJEB47105 Whole genome sequencing of SARS-CoV-2 in Estonia PRJEB47120 Whole genome sequencing of SARS-CoV-2 in Estonia PRJEB47598 Whole genome sequencing of SARS-CoV-2 in Estonia PRJEB48077 Whole genome sequencing of SARS-CoV-2 in Estonia PRJEB48558 Whole genome sequencing of SARS-CoV-2 in Estonia PRJEB48886 Whole genome sequencing of SARS-CoV-2 in Estonia PRJEB50528 Whole genome sequencing of SARS-CoV-2 in Estonia PRJEB50535 Whole genome sequencing of SARS-CoV-2 in Estonia PRJEB50546 Whole genome sequencing of SARS-CoV-2 in Estonia PRJEB50677 Whole genome sequencing of SARS-CoV-2 in Estonia PRJEB47110 Whole genome sequencing of SARS-CoV-2 in Estonia PRJEB47180 Whole genome sequencing of SARS-CoV-2 in Estonia PRJEB47183 Whole genome sequencing of SARS-CoV-2 in Estonia PRJEB47194 Whole genome sequencing of SARS-CoV-2 in Estonia PRJEB47585 Whole genome sequencing of SARS-CoV-2 in Estonia PRJEB47873 Whole genome sequencing of SARS-CoV-2 in Estonia PRJEB48696 Whole genome sequencing of SARS-CoV-2 in Estonia PRJEB48711 Whole genome sequencing of SARS-CoV-2 in Estonia PRJEB50312 Whole genome sequencing of SARS-CoV-2 in Estonia PRJEB50486 Whole genome sequencing of SARS-CoV-2 in Estonia PRJEB50540 Whole genome sequencing of SARS-CoV-2 in Estonia PRJEB50559 Whole genome sequencing of SARS-CoV-2 in Estonia PRJEB47121 Whole genome

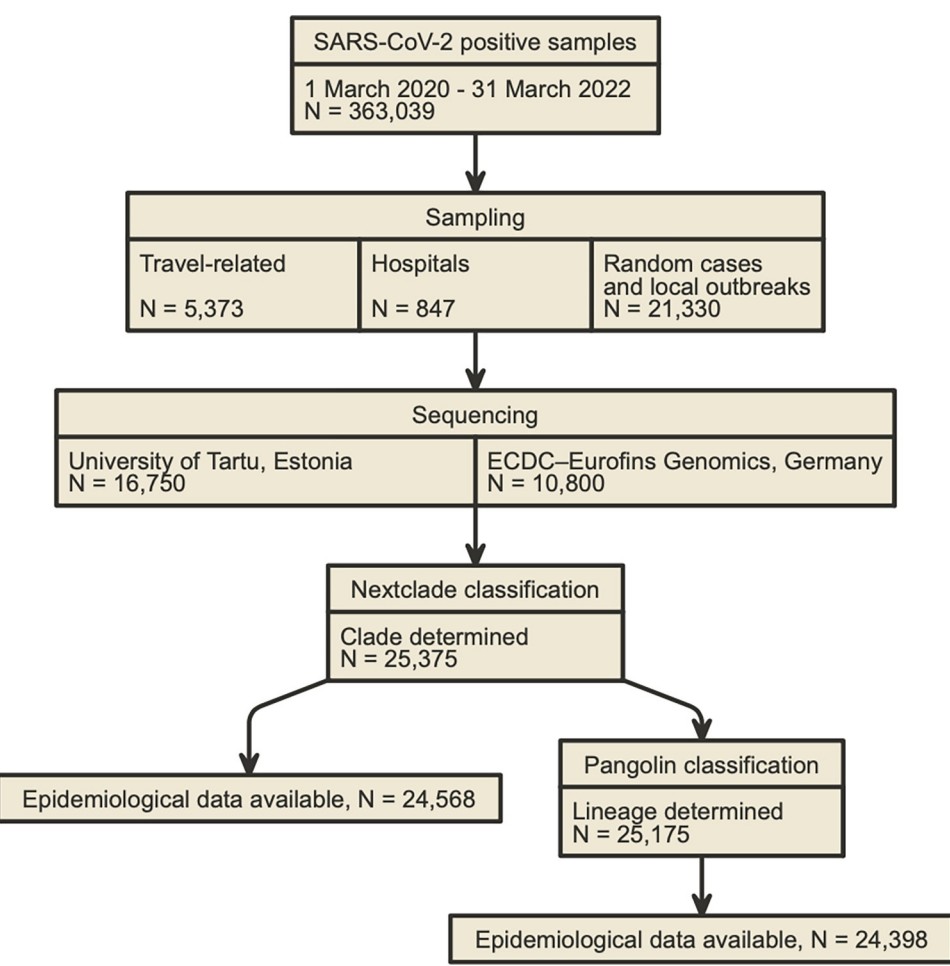

**Fig 1. The workflow for the study population.** Epidemiological data availability means that, at least, the date of sampling and age or sex are present. The use of 'ECDC' refers to the European Centre for Disease Prevention and Control in Sweden.

hospitals, at dates between 13 March 2020 and 31 March 2022 as part of the national SARS-CoV-2 PCR testing process. To be able to select samples for genotyping purposes, various sampling strategies were applied according to public health needs in Estonia. During the first wave in 2020, samples were sequenced from the first cases and local SARS-CoV-2 outbreaks. After the first wave, sequencing was carried out by using randomly-selected samples (median 336 samples per week). In addition, targeted samples were occasionally sequenced from hospitalised individuals, regardless of the reason for hospitalisation, and from those who had a history of international travel in the 14 days prior to their producing a positive PCR test for SARS-CoV-2 (Fig 1). Sample genotyping data was last accessed on 11 January 2023.

## SARS-CoV-2 sequencing

During the study period two different protocols were used for sequencing. Between March 2020 and January 2021, RNA was reverse-transcribed and amplified in 2.5kb products [8]. Pooled amplicons of each sample were used to prepare NGS libraries with Nextera XT and were clustered with MiSeq Reagent Kit v2 (500-cycles) on 2 x 150-cycle paired-end runs (Illumina Inc, San Diego, CA, USA). Since February 2021, an Illumina COVIDSeq Test Kit Artic

sequencing of SARS-CoV-2 in Estonia PRJEB48640 Whole genome sequencing of SARS-CoV-2 in Estonia PRJEB48772 Whole genome sequencing of SARS-CoV-2 in Estonia PRJEB49981 Whole genome sequencing of SARS-CoV-2 in Estonia PRJEB50007 Whole genome sequencing of SARS-CoV-2 in Estonia PRJEB50859 Whole genome sequencing of SARS-CoV-2 in Estonia PRJEB47123 Whole genome sequencing of SARS-CoV-2 in Estonia PRJEB49812 Whole genome sequencing of SARS-CoV-2 in Estonia PRJEB50186 Whole genome sequencing of SARS-CoV-2 in Estonia PRJEB47125 Whole genome sequencing of SARS-CoV-2 in Estonia PRJEB48515 Whole genome sequencing of SARS-CoV-2 in Estonia PRJEB49070 Whole genome sequencing of SARS-CoV-2 in Estonia PRJEB49805 Whole genome sequencing of SARS-CoV-2 in Estonia PRJEB49810 Whole genome sequencing of SARS-CoV-2 in Estonia PRJEB50341 Whole genome sequencing of SARS-CoV-2 in Estonia PRJEB50524 Whole genome sequencing of SARS-CoV-2 in Estonia PRJEB50525 Whole genome sequencing of SARS-CoV-2 in Estonia PRJEB50531 Whole genome sequencing of SARS-CoV-2 in Estonia PRJEB50675 Whole genome sequencing of SARS-CoV-2 in Estonia PRJEB50997 Whole genome sequencing of SARS-CoV-2 in Estonia PRJEB47101 Whole genome sequencing of SARS-CoV-2 in Estonia PRJEB47102 Whole genome sequencing of SARS-CoV-2 in Estonia PRJEB47992 Whole genome sequencing of SARS-CoV-2 in Estonia PRJEB48559 Whole genome sequencing of SARS-CoV-2 in Estonia PRJEB49076 Whole genome sequencing of SARS-CoV-2 in Estonia PRJEB49978 Whole genome sequencing of SARS-CoV-2 in Estonia PRJEB50886 Whole genome sequencing of SARS-CoV-2 in Estonia PRJEB47116 Whole genome sequencing of SARS-CoV-2 in Estonia PRJEB47118 Whole genome sequencing of SARS-CoV-2 in Estonia PRJEB47190 Whole genome sequencing of SARS-CoV-2 in Estonia PRJEB47871 Whole genome sequencing of SARS-CoV-2 in Estonia PRJEB48440 Whole genome sequencing of SARS-CoV-2 in Estonia PRJEB48701 Whole genome sequencing of SARS-CoV-2 in Estonia PRJEB48771 Whole genome sequencing of SARS-CoV-2 in Estonia PRJEB48873 Whole genome sequencing of SARS-CoV-2 in Estonia PRJEB49167 Whole genome sequencing of SARS-CoV-2 in Estonia PRJEB49807 Whole genome sequencing of SARS-CoV-2 in Estonia PRJEB50311 Whole genome sequencing of SARS-CoV-2 in Estonia PRJEB50327 Whole genome

v3 or v4/v4.1 (Illumina Inc, San Diego, CA, USA) was used for reverse transcribing, followed by amplification and sequencing by the Illumina NextSeq500 using paired-end 75bp or 150bp reads. A proportion of samples were sequenced by Eurofins Genomics Germany GmbH, brokered by the ECDC. Further details of the sequencing are described in the 'Methods' section in the S1 Appendix. Raw sequence data for analysed samples were submitted to the European Nucleotide Archive (ENA) using the ENA upload tool for Galaxy (https://github.com/usegalaxy-eu/ena-upload-cli). Raw sequences are deposited under ENA Project accessions as shown in Table 2 in the S2 Appendix.

## Data collection

Epidemiological and clinical data were retrieved from the Estonian Health Board (https://www.terviseamet.ee/en) and the Estonian Health and Welfare Information Systems Centre (https://www.tehik.ee/en). The following data were collected: sampling date, district of origin, sex, age, possible source of infection, history of visited countries within the 14 days prior to diagnosis, a history of SARS-CoV-2 vaccination, and a history of hospitalisation due to COVID-19 (duration, stay in an intensive care unit, the need for mechanical ventilation, and any reason for discharge). Epidemiological and clinical data was last accessed on 11 January 2023. For this study, the authors had no access to any information which could be used to identify individual participants during or after data collection.

## Statistical analysis

All data analyses were conducted using R v4.2.1 (2022-06-23). The study population baseline characteristics table was generated by using the gtsummary R package v1.6.3 [9]. Bayesian modelling was carried out using the R libraries, rstan v2.21.7 [10] and brms v2.18.0 [11]. Weakly informative priors were used to fit the models, and a minimum of 2,000 iterations, including warm-up iterations which amounted to half of all iterations, and three to four chains were used to fit the models. The work of extracting, summarising, and visualising draws from brms models was carried out with the tidybayes v3.0.2 [12] and emmeans v1.8.1–1 [13] R packages. The Shannon index (H) was calculated by using the vegan R package v2.6–2 [14] as $H = -\Sigma[(p_i) * \log(p_i)]$, where $p_i$ is the proportion of the $i^{th}$ species in an entire community. In terms of diversity analyses, the species was defined as a distinct Pango lineage. The evenness index (J) was calculated as $J = H / \log(S)$, where 'H' is the Shannon index and 'S' is the observed number of Pango lineages. Raw data manipulations, including importation, transformation, and summaries, were generated with the tidyverse R package v1.3.1 [15]. Plots were prepared using the ggplot2 R package v3.3.6 [16].

## Ethics statement

The study was approved by the Research Ethics Committee at the University of Tartu [approvals 304/T-1, 324/T-1]. Informed consent was waived according to § 6 of the national 'Personal Data Protection Act'.

## Results

We obtained a total of 25,375 (or 92%) complete or near-complete SARS-CoV-2 genomes, representing approximately 7% of the PCR-confirmed SARS-CoV-2 infections in Estonia during the study period (Fig 1). Our dataset covers more than 5% of SARS-CoV-2 PCR-confirmed tests for 72% of the weeks during the study period (Fig 2).

sequencing of SARS-CoV-2 in Estonia PRJEB50436 Whole genome sequencing of SARS-CoV-2 in Estonia PRJEB50678 Whole genome sequencing of SARS-CoV-2 in Estonia PRJEB49988 Whole genome sequencing of SARS-CoV-2 in Estonia PRJEB50316 Whole genome sequencing of SARS-CoV-2 in Estonia PRJEB50526 Whole genome sequencing of SARS-CoV-2 in Estonia PRJEB50549 Whole genome sequencing of SARS-CoV-2 in Estonia PRJEB47100 Whole genome sequencing of SARS-CoV-2 in Estonia PRJEB47103 Whole genome sequencing of SARS-CoV-2 in Estonia PRJEB47104 Whole genome sequencing of SARS-CoV-2 in Estonia PRJEB47113 Whole genome sequencing of SARS-CoV-2 in Estonia PRJEB47114 Whole genome sequencing of SARS-CoV-2 in Estonia PRJEB47115 Whole genome sequencing of SARS-CoV-2 in Estonia PRJEB47119 Whole genome sequencing of SARS-CoV-2 in Estonia PRJEB47145 Whole genome sequencing of SARS-CoV-2 in Estonia PRJEB47227 Whole genome sequencing of SARS-CoV-2 in Estonia PRJEB47674 Whole genome sequencing of SARS-CoV-2 in Estonia PRJEB48429 Whole genome sequencing of SARS-CoV-2 in Estonia PRJEB50189 Whole genome sequencing of SARS-CoV-2 in Estonia PRJEB50192 Whole genome sequencing of SARS-CoV-2 in Estonia PRJEB50680 Whole genome sequencing of SARS-CoV-2 in Estonia PRJEB50860 Whole genome sequencing of SARS-CoV-2 in Estonia PRJEB52934 Whole genome sequencing of SARS-CoV-2 in Estonia PRJEB52588 Whole genome sequencing of SARS-CoV-2 in Estonia PRJEB52841 Whole genome sequencing of SARS-CoV-2 in Estonia.

**Funding:** This work was funded by the Ministry of Education and Research, Republic of Estonia [contract number 1.1-6.2/21/298] and by the University of Tartu. This work was funded partially by the European Union through HORIZON Coordination and Support Actions [grant agreement 101079349] "Boosting the One Health Research Excellence and Management Capacity of the Estonian University of Life Sciences". Views and opinions expressed are however those of the author(s) only and do not necessarily reflect those of the European Union or European Health and Digital Executive Agency. Neither the European Union nor the granting authority can be held responsible for them. This work was funded partially by the European Union through EU4Health Programme (EU4H) [grant agreement 101102733] "Delivering a Unified Research Alliance of Biomedical and public health Laboratories against Epidemics". Views and opinions expressed are

## Study population

Table 1 presents the characteristics of the study population as was stratified by the VOC waves. Overall, our study population's sex ratio and age group distribution match up well with the population of SARS-CoV-2 RT-PCR-positive people in Estonia. Binomial proportion analysis showed that the study population consisted of 1.6–2.8 percentage points more females than males while, in the target population involving SARS-CoV-2 RT-PCR-positive people in Estonia, females were even more prevalent than males (4.1–4.3 percentage points). The median age of individuals (37 years of age) was similar both in the study and in the target population. Age groups between twenty and sixty years were over-represented relative to their prevalence in the entire Estonian population (S1A Fig). Most people who presented with COVID-19 symptoms at the time of testing (96%) and about two-thirds of the people overall (66%) were infected in Estonia. About 3% of SARS-CoV-2 RT-PCR-positive people were hospitalised. During the study period the prevalence of people in the below-twenty year-old age group increased, while the prevalence of people aged between 60–79 decreased (both trends have a posterior probability of >0.999). No substantial changes were observed in the prevalence of other age groups (S1B Fig). The sex ratio showed a substantial decrease in the proportion of males in the under-twenty group and the 20–39 age range (with a posterior probability of 0.995 and 0.999 respectively), with no changes in age groups above those ages (see S2 Fig).

## The prevalence of clades and lineages during the VOC waves

Overall, 19 Nextstrain clades and 199 Pango lineages were identified during the study period. Those variants which belonged to the Delta (43%), Omicron (27%), and Alpha (23%) VOCs were the most highly-abundant of all sequenced SARS-CoV-2 genomes (Table 1 and S1 Table). During the pre-VOC period in 2020, the two most prevalent clades were 20B (51%), and 20A (42%) (Fig 3A). The Alpha VOC (20I) was first identified in Estonia on 31 December 2020 in a sample which had been collected from an individual with travel history. Additional cases of the Alpha VOC were identified in January 2021, followed by the local spread of the variant which eventually led to the Alpha wave, from W9 to W22 in 2021. Clades which belonged to the Delta VOC emerged thereafter and out-competed the Alpha VOC by W26 in 2021 (Fig 3A and 3B). During the Delta wave we detected three clades (21A, 21I, and 21J), with 21J emerging as dominant. In contrast to the Alpha and Delta waves, the Omicron VOC resulted in two consecutive waves, first with 21K (BA.1 and its sub-lineages) from W52 in 2021 to W5 in 2022, and then with 21L (BA.2 and its sub-lineages), starting in W7 in 2022 (Fig 3A and 3B). We detected a total of twenty-seven different lineages in 2020, with the three most highly-prevalent lineages being B.1.1 (28%), B.1.1.10 (18%), and B.1.258 (13%). The Alpha wave in 2021 was characterised by the dominance of the B.1.1.7 lineage (99%) whereas, during the Delta wave, we detected a total of eighty-two different Delta VOC lineages and sub-lineages, with AY.122 being predominant (43%), followed by AY.100 (10%) and AY.43 (7.3%). During the two Omicron waves up to 31 March 2022, we detected twenty-eight BA.1 and 16 BA.2 sub-lineages (see S1 Table).

## The association between travel-related cases and VOC

We found that the proportion of travel-related cases was about 20% in populations which had become infected with the Alpha-, Delta VOC, or other variants, and about 40–50% in those who had become infected with Omicron or Beta VOC (S3 Fig). These cases were associated with 88 different countries. The top five countries into which VOCs had mainly been imported included the neighbouring countries around Estonia (Finland, Sweden, and Russia), and popular travel destinations (Egypt and Great Britain). The Beta VOC import was related to a very

however those of the author(s) only and do not necessarily reflect those of the European Union or European Health and Digital Executive Agency. Neither the European Union nor the granting authority can be held responsible for them. The funders had no role in study design, data collection and analysis, decision to publish, or preparation of the manuscript.

**Competing interests:** The authors have declared that no competing interests exist.

limited set of countries and did not result in a wave in Estonia. Compared to the Beta, the Omicron VOC displayed a wider geography, one which was similar to that of the Alpha and Delta VOCs (see S3C Fig).

## Higher diversity amongst travel-related cases

We identified a total of 165 different Pango lineages from the period for which we had sample origin info (starting from 7 Feb 2021). A total of 57 (or 35%) of those were unique to individuals who reported that they had travelled compared to 23 (13%) lineages which were uniquely assignable to domestic infections, while 86 (52%) lineages were present in both groups. Out of those 86 lineages which were found in both groups, a total of 51 (59%) were first identified in individuals who had reported that they had travelled. The Shannon diversity index of Pango lineages decreased steeply during the Alpha and Omicron waves, something which was in contrast with the Delta wave in which diversity rebounded quickly after the Alpha wave and remained at a relatively higher level until the end of the wave (Fig 4A). Diversity displayed small peaks at the beginning of waves and was higher amongst travel-related cases when compared to domestic cases in about 68% of weeks during the Alpha, Delta, and Omicron waves (Fig 4B). The second half of the Delta wave was characterised by higher levels of diversity in domestic cases. The observed number of lineages was higher during the Delta wave and the first Omicron wave (21K, BA.1 and its sub-lineages) when compared to the Alpha wave and

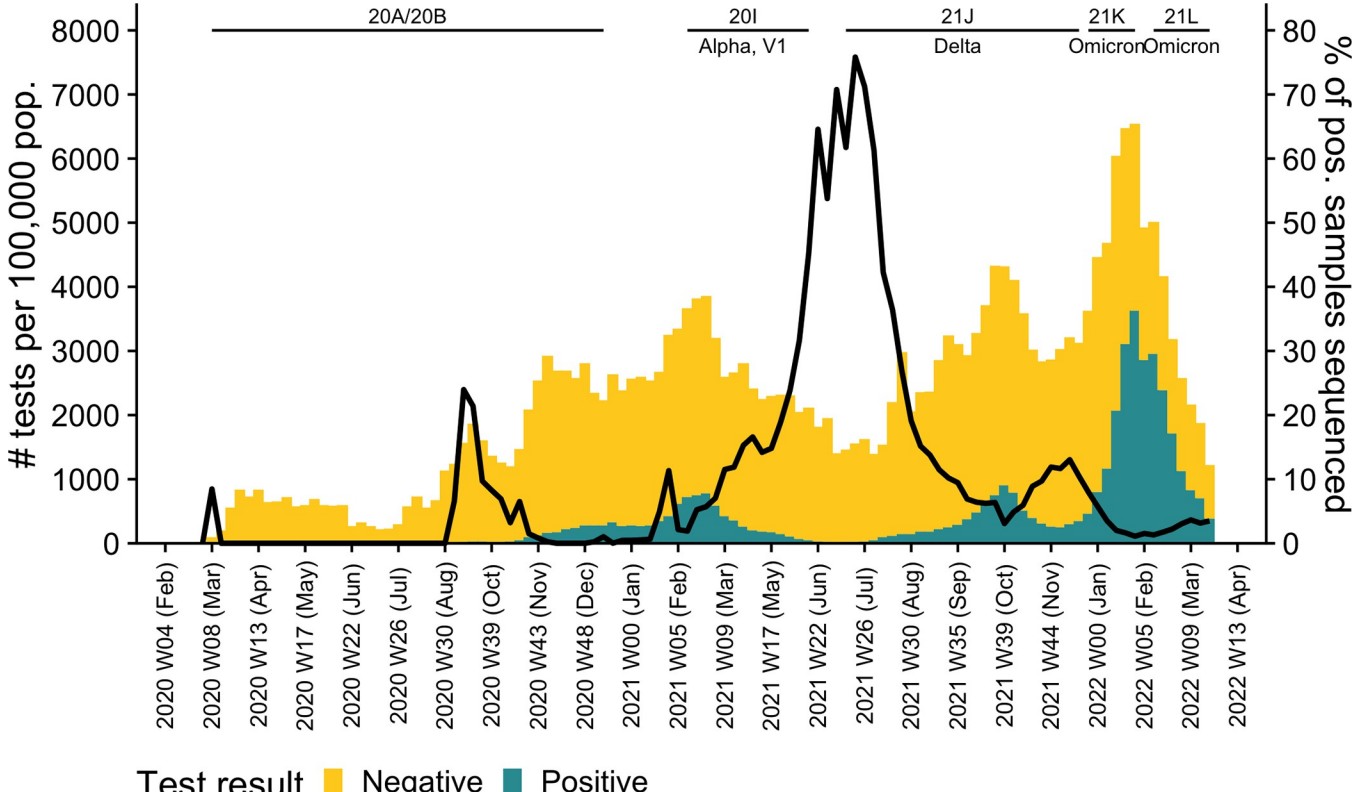

**Fig 2. Testing and sequencing SARS-CoV-2 in Estonia.** The bars show the number of tests which were carried out (left axis) and the line denotes the percentage of positive samples sequenced (right axis). Vertical lines denote the duration of waves, defined by the prevalence of Nextstrain clade(s). The wave start and end were defined as a week in which the lower or upper bound respectively of the 95% confidence interval of clade prevalence, as obtained through the Clopper-Pearson method, crossed the 50% threshold. SARS-CoV-2 test results were downloaded from the Estonian COVID-19 open-data portal (https://opendata.digilugu.ee; last accessed 5 March 2024).

**Table 1. The characteristics of the study population.**

| Variable | Total SARS-CoV-2 positive samples N = 363,039 | N | Overall, N = 25,375 | Alpha, N = 5,842 | Delta, N = 10,793 | Omicron, N = 6,851 | Other, N = 1,806 | Other VOC, N = 83 |
|---|---|---|---|---|---|---|---|---|
| | | | SARS-CoV-2 positive samples sequenced | | | | | |
| **Sex, n (%)** | | 24,344 | | | | | | |
| Female | 194,068 (54) | | 12,712 (52) | 2,942 (51) | 5,581 (52) | 3,384 (54) | 771 (51) | 34 (41) |
| Male | 168,660 (46) | | 11,632 (48) | 2,860 (49) | 5,133 (48) | 2,849 (46) | 742 (49) | 48 (59) |
| Unknown | 311 | | 1,031 | 40 | 79 | 618 | 293 | 1 |
| **Age, median (IQR)** | 37 [20, 52] | 24,567 | 37 (24, 52) | 40 (26, 55) | 36 (23, 50) | 37 (23, 50) | 42 (27, 57) | 37 (28, 49) |
| Unknown | 2,240 | | 808 | 23 | 64 | 432 | 289 | 0 |
| **Age, n (%)** | | 24,567 | | | | | | |
| 0–19 | 88,940 (25) | | 4,615 (19) | 930 (16) | 2,205 (21) | 1,249 (19) | 221 (15) | 10 (12) |
| 20–39 | 112,361 (31) | | 8,709 (35) | 1,966 (34) | 3,874 (36) | 2,349 (37) | 487 (32) | 33 (40) |
| 40–59 | 102,343 (28) | | 7,526 (31) | 1,887 (32) | 3,201 (30) | 1,922 (30) | 481 (32) | 35 (42) |
| 60–79 | 46,138 (13) | | 3,090 (13) | 882 (15) | 1,202 (11) | 725 (11) | 276 (18) | 5 (6) |
| >80 | 11,017 (3) | | 627 (3) | 154 (3) | 247 (2) | 174 (3) | 52 (3) | 0 (0) |
| Unknown | 2,240 | | 808 | 23 | 64 | 432 | 289 | 0 |
| **Possible place of contracting the infection, n (%)** | | 14,341 | | | | | | |
| Abroad/related to travelling | | | 4,912 (34) | 715 (21) | 2,491 (32) | 1,446 (60) | 227 (31) | 33 (55) |
| Domestic | | | 9,429 (66) | 2,692 (79) | 5,231 (68) | 974 (40) | 505 (69) | 27 (45) |
| Unknown | | | 11,034 | 2,435 | 3,071 | 4,431 | 1,074 | 23 |
| **Presenting with symptoms, n (%)** | | 14,542 | | | | | | |
| No | | | 632 (4) | 160 (6) | 339 (4) | 82 (4) | 50 (8) | 1 (6) |
| Yes | | | 13,910 (96) | 2,340 (94) | 9,026 (96) | 1,980 (96) | 548 (92) | 16 (94) |
| Unknown | | | 10,833 | 3,342 | 1,428 | 4,789 | 1,208 | 66 |
| **Covid Cq, mean (SD)** | | 4,135 | 16 (4) | NA (NA) | 16 (5) | 16 (4) | NA (NA) | NA (NA) |
| Unknown | | | 21,240 | 5,842 | 10,774 | 2,735 | 1,806 | 83 |
| **Average Cq, mean (SD)** | | 17,843 | 17 (5) | 18 (6) | 17 (5) | 18 (5) | 20 (6) | 19 (6) |
| Unknown | | | 7,532 | 895 | 219 | 5,431 | 978 | 9 |
| **Vaccination, n (%)** | | 24,233 | | | | | | |
| Boosted | | | 1,065 (4) | 0 (0) | 70 (1) | 995 (16) | 0 (0) | 0 (0) |
| Fully vaccinated | | | 6,489 (27) | 222 (4) | 3,684 (34) | 2,563 (41) | 20 (1) | 0 (0) |
| Unvaccinated | | | 16,679 (69) | 5,568 (96) | 6,959 (65) | 2,692 (43) | 1,379 (99) | 81 (100) |
| Unknown | | | 1,142 | 52 | 80 | 601 | 407 | 2 |
| **Hospitalisation, n (%)** | | 24,233 | | | | | | |
| No | | | 23,498 (97) | 5,481 (95) | 10,413 (97) | 6,214 (99) | 1,309 (94) | 81 (100) |
| Yes | | | 735 (3) | 309 (5) | 300 (3) | 36 (1) | 90 (6) | 0 (0) |
| Unknown | | | 1,142 | 52 | 80 | 601 | 407 | 2 |
| **Days in COVID unit, median (IQR)** | | 723 | 7 (5, 12) | 8 (5, 12) | 7 (5, 11) | 4 (2, 10) | 10 (6, 12) | NA (NA, NA) |
| Unknown | | | 24,652 | 5,542 | 10,496 | 6,815 | 1,716 | 83 |
| **Days in intensive care, median (IQR)** | | 67 | 9 (5, 16) | 8 (3, 14) | 9 (7, 15) | 1 (1, 1) | 12 (6, 21) | NA (NA, NA) |
| Unknown | | | 25,308 | 5,816 | 10,761 | 6,850 | 1,798 | 83 |
| **Days on ventilation, median (IQR)** | | 47 | 9 (5, 15) | 11 (5, 20) | 9 (5, 14) | NA (NA, NA) | 8 (6, 12) | NA (NA, NA) |
| Unknown | | | 25,328 | 5,827 | 10,768 | 6,851 | 1,799 | 83 |
| **Reason of discharge from hospital, n (%)** | | 735 | | | | | | |

*(Continued)*

**Table 1.** (Continued)

| Variable | Total SARS-CoV-2 positive samples | | SARS-CoV-2 positive samples sequenced | | | | | |
|---|---|---|---|---|---|---|---|---|
| | N = 363,039 | N | Overall, N = 25,375 | Alpha, N = 5,842 | Delta, N = 10,793 | Omicron, N = 6,851 | Other, N = 1,806 | Other VOC, N = 83 |
| Death | | | 78 (11) | 44 (14) | 23 (8) | 2 (6) | 9 (10) | |
| Recovery | | | 470 (64) | 199 (64) | 196 (65) | 28 (78) | 47 (52) | |
| Transfer to non-COVID unit | | | 184 (25) | 64 (21) | 80 (27) | 6 (17) | 34 (38) | |
| Unknown | | | 3 (0) | 2 (1) | 1 (0) | 0 (0) | 0 (0) | |
| **Places of domestic infections, n (%)** | | 9,429 | | | | | | |
| Educational or children's facility | | | 721 (8) | 112 (4) | 467 (9) | 120 (12) | 21 (4) | 1 (4) |
| Home/Family | | | 5,342 (57) | 1,568 (58) | 2,905 (56) | 614 (63) | 236 (47) | 19 (70) |
| Other | | | 1,329 (14) | 304 (11) | 815 (16) | 98 (10) | 111 (22) | 1 (4) |
| Visiting acquaintances | | | 650 (7) | 152 (6) | 418 (8) | 74 (8) | 6 (1) | 0 (0) |
| Workplace | | | 1,387 (15) | 556 (21) | 626 (12) | 68 (7) | 131 (26) | 6 (22) |

Displayed in the table are the epidemiological and clinical characteristics of populations which have been infected with SARS-CoV-2 VOCs in Estonia between March 2020 and March 2022. Other VOCs consist of Beta and Gamma VOCs. See S1 Table for lineages which belong to strata. IQR, interquartile range; NA, not applicable; SD, standard deviation. The dimension of values given in parentheses is shown in the 'Variable' column.

the second Omicron wave both in travel-related and domestic cases (21L, BA.2 and its sub-lineages) (S5A Fig). For most of the weeks under consideration, the number of unique lineages was higher amongst travel-related cases (S5B Fig). The weekly number of unique lineages in travel-related cases displayed a substantial peak relative to domestically circulating lineages at the beginning of the first Omicron wave (S5A and S5B Fig). The evenness index sharply decreased during the Alpha wave when 99% of cases belonged to the B.1.1.7 lineage (see S1 Table), but this rebounded before the Delta wave, and remained relatively stable afterwards with no single lineage becoming prevalent (S5C Fig). The evenness index was higher in imported cases in about 66% of overall weeks during the Alpha, Delta, and Omicron waves (S5D Fig).

### The association of RT-PCR Cq values with VOC

We observed that people who had become infected with the Delta VOC had lower average RT-PCR Cq values of the ORF1a, S gene, and N gene than people who had been infected with the Alpha or Omicron (16.6 versus 18.2 or 18.2 respectively) VOC (Table 1). Two of the Delta clades (21I and 21J) displayed considerably lower average RT-PCR Cq values (16.0, 95% credible interval (CI) [15.4, 16.8] and 16.8, 95% CI [16.2, 17.3] respectively), adjusted for age and vaccination status, than all other clades, such as 21K Omicron (18.7, 95% CI [17.8, 19.2]) or 20I Alpha (18.9 95% CI [18.5, 19.3]) (see S4 Fig). The 21A Delta clade displayed higher Cq values than did all other clades (except EU1) (22.5, 95% CI [21.2, 23.8]).

### The association between hospitalisation and clade and vaccination

We found that the probability of hospitalisation was strongly related to age and COVID-19 vaccination status (Fig 5A). Overall, the probability of hospitalisation was very low for people who were under forty years of age but started to increase after that, whereas vaccinated people had a lower probability of being hospitalised when compared to unvaccinated people. The beneficial effect of any vaccination increased considerably with age, as the odds of hospitalisation

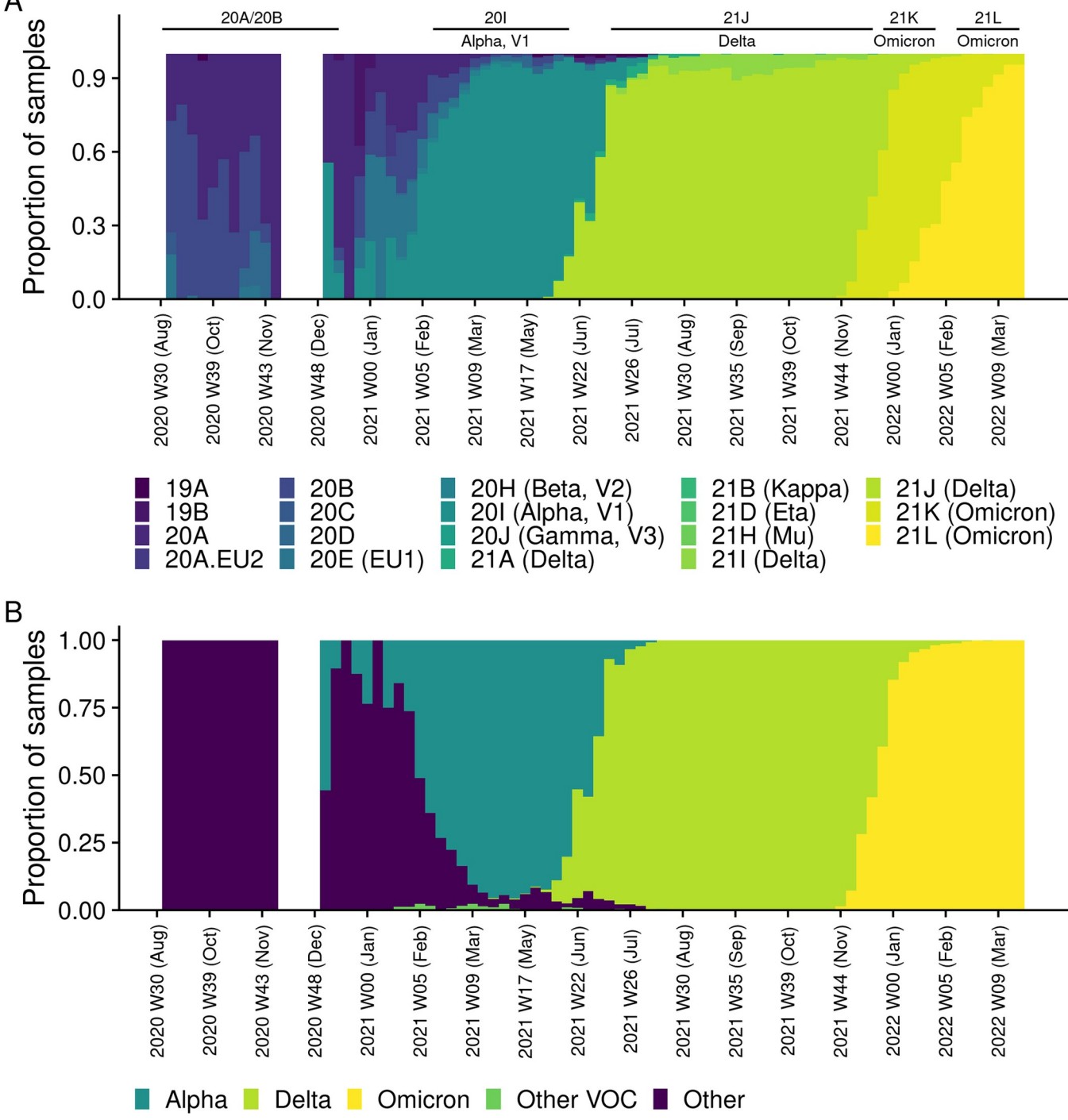

**Fig 3. Waves of SARS-CoV-2 in Estonia.** (A) the prevalence of Nextstrain clades; (B) the prevalence of VOCs. Full-length sequencing was not applied to samples collected between W47 to W52 in 2020.

decreased with age in the vaccinated population when compared to the unvaccinated (OR = 0.71, 95% CI [0.58, 0.84] at forty years of age; OR = 0.37, 95% CI [0.3, 0.45] at sixty, and OR = 0.19, 95% CI [0.13, 0.27] at eighty. This association between age and vaccination effect was similar across all common clades (Fig 5B). The probability of hospitalisation was higher

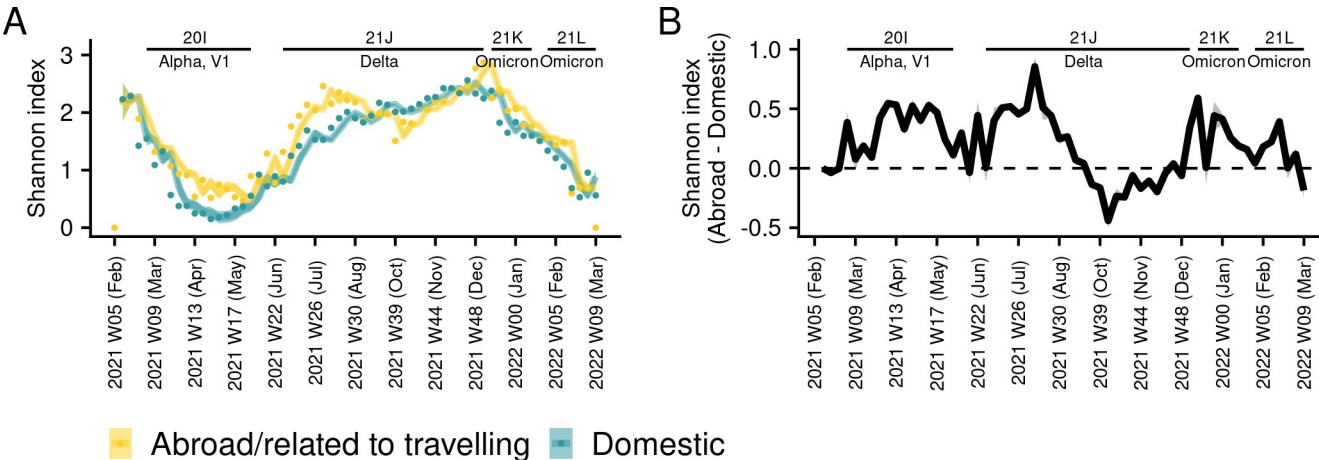

**Fig 4. The diversity of SARS-CoV-2 lineages in domestic and travel-related cases in Estonia.** (A) Shannon diversity index. Points denote individual weekly observations. The line denotes the autoregressive model which has been fitted to the data, and the shaded ribbon denotes a 95% credible interval, N = 14,341; (B) the effect-size of the Shannon index, imported cases compared to domestic cases. The line denotes the effect size as derived from the model fit which is shown in Panel A, and the shaded ribbon denotes a 95% credible interval. The model summary is presented in Table 3 in the S2 Appendix.

for males over fifty than it was for females of the same age regardless of vaccination status (S6A Fig). The odds of hospitalisation for males when compared to females was, for fifty year-olds, OR = 1.35, 95% CI [1.17, 1.54] and, for seventy-five year-olds, OR = 2.39, 95% CI: [1.85, 2.99] (S6B Fig). Pairwise comparisons of the clades showed that 21K and 21L Omicron-infected people had a lower probability of hospitalisation in age groups which started at the age of forty, independent of any vaccination status when compared to people of the same age who had been infected with any other common clades (Fig 5C and S7 and S8 Figs). In the population of twenty year-olds and below, 21K and 21L Omicron-infected people had a similar hospitalisation level of probability when compared to other clades (Fig 5C and S7 and S8 Figs). Pairwise comparisons of clades other than 21K and 21L Omicron showed a similar probability of hospitalisation (see S7 and S8 Figs for all pairwise comparisons).

## Discussion

In this countrywide study, we characterise the circulating variants in Estonia between March 2020 and March 2022 by genotyping about 7% of SARS-CoV-2-positive samples.

We demonstrate the following:

1) the prevalence of the clades in Estonia resembled those in other European countries

2) different VOC waves varied in terms of Pango lineage diversity, with the highest levels of diversity being seen in the Delta wave and the lowest in the Alpha wave

3) the probability of hospitalisation was associated with clade, as two Omicron clades (21K and 21L) displayed lower hospitalisation rates when compared to all other common clades

4) the protective effect of the vaccination against severe disease, with such disease being associated with hospitalisation, was seen in individuals who were aged forty and above, with the effect strongly increasing with age and being independent of the involved clade.

### Changes in the most highly-affected population groups during the pandemic

The genotypic surveillance of SARS-CoV-2 in Estonia presents trends which generally match those of the rest of Europe, being and is characterised by the prevalence of clades 20A and 20B

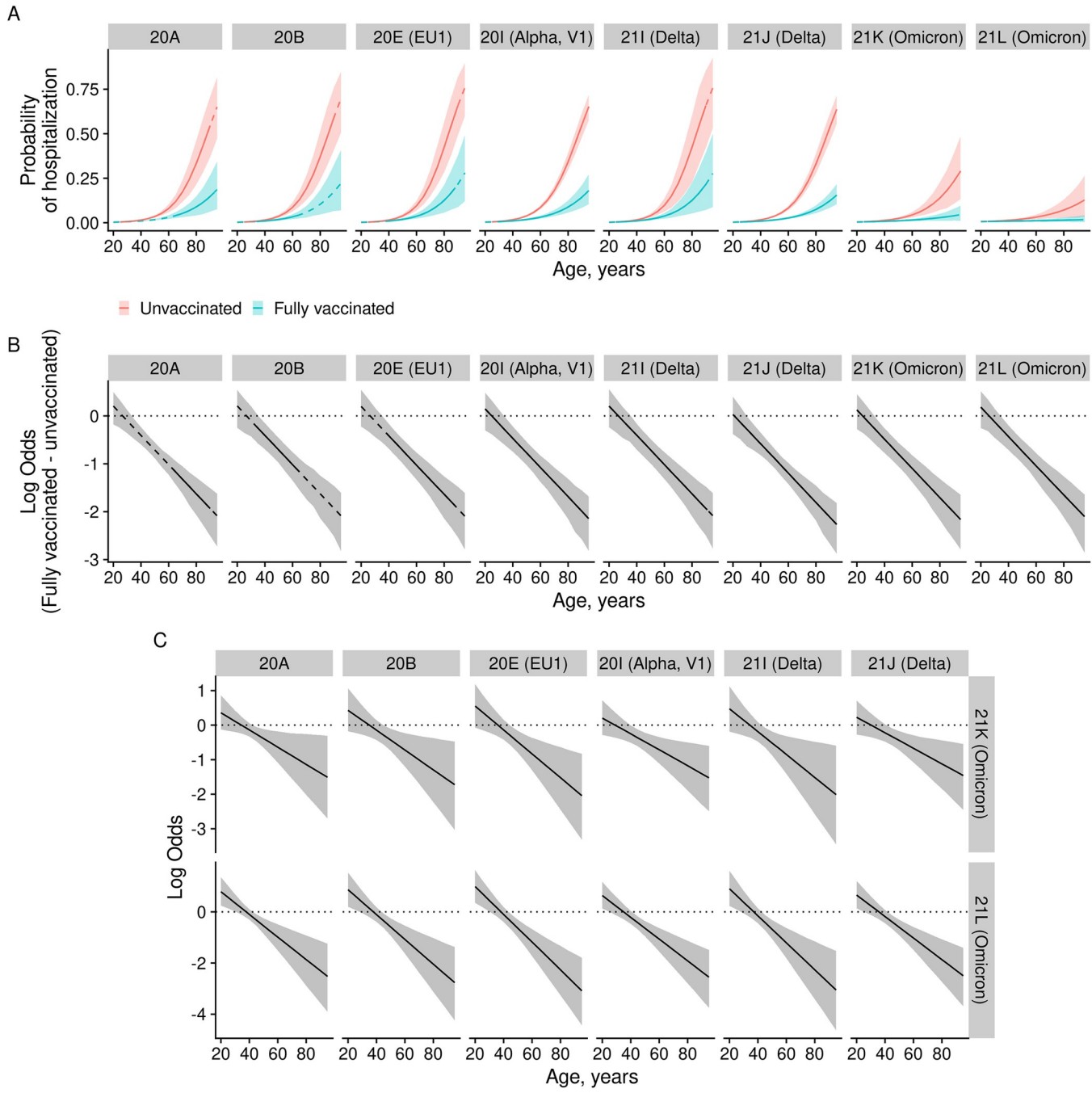

**Fig 5. The hospitalisation of SARS-CoV-2-positive individuals in Estonia.** (A) the probability of hospitalisation is associated with vaccination status, age, and clade. Posterior summaries of the conditional effect of the clade, vaccination status, and age in logistic regression when adjusted for sex, the number of weekly cases, and population vaccination coverage, N = 23,456; (B) log the odds ratio of hospitalisation in vaccinated individuals when compared to unvaccinated SARS-CoV-2-positive individuals; (C) log the odds ratio in terms of the hospitalisation of unvaccinated SARS-CoV-2-positive individuals who have been infected with Omicron 21K (upper panels) or 21L (lower panels) when compared to individuals who have been infected with other common SARS-CoV-2 clades. The line denotes the model's best estimate, and the ribbon denotes a 95% credible interval. The solid line denotes the span of ages which have been observed for each clade, while the dashed line denotes unobserved ages implied by the underlying model. The dotted line denotes log odds = 0. Panels B and C are based on the same model as for A, while the model summary is presented in Table 4 in the S2 Appendix.

in 2020, and then in 2021–2022 by three successive waves of SARS-CoV-2 which were dominated by Alpha, Delta, and Omicron VOC in that order (https://www.ecdc.europa.eu/en/covid-19). As for those population subgroups which were affected, as with others [17], we observed that individuals who were aged between 20–59 years were over-represented in Estonia (>30%) relative to their proportion in the general population (26%). Again as with others [18, 19], we observed an age-group prevalence shift from individuals who were aged forty and above during the first waves of the pandemic to individuals who were below forty, with the most substantial increase being seen during the Omicron waves in individuals who were aged nineteen and below. We could speculate that such shifts can be attributed to several measured and non-measured factors, such as asymptomatic infections and/or milder symptoms in testing younger people, the early vaccination of older at-risk age groups when compared to people in the younger age groups, and public health efforts to manage the pandemic, to mitigation measure enforcement, and also to comply with such measures. The introduction of antigen quick-testing in schools in November 2021 may also have substantially contributed to the measured increase in the prevalence of under-nineteens showing an increase in infections.

## Lineage diversity during the waves

Lineage diversity in terms of the Alpha wave was distinct from that of all other waves, including the pre-VOC wave, and the Delta and Omicron waves, through the overwhelming dominance of one single lineage: B.1.1.7. The B.1.1.7 lineage was more highly-transmissible than its pre-VOC predecessors, resulting in its worldwide spread, and this became dominant in a large number of countries [20–22]. Omicron waves displayed a similar pattern to Alpha within our study timeframe, albeit with a substantial decrease in diversity. Diversity was at its highest during the Delta wave and, in contrast to the Alpha and Omicron waves, remained high throughout the wave. We could speculate that higher Delta diversity was associated with higher virus loads, as Delta infections were shown to carry about six times more viral RNA when compared to Alpha [23], something which is supported by our findings for the two Delta clades which displayed the lowest Cq values. Interestingly, we observed an apparent increase in the diversity of the imported variants before or at the beginning of new waves. Variants which were present only in those individuals who reported that they had recently travelled and variants which were first detected in individuals who had reported travelling represent 70% of all variants which were circulating in Estonia during the same period, suggesting 70% efficiency in intercepting new variants entering the country.

## Hospitalisation

As is the case with others [24, 25], our analysis of hospitalisation data suggests that the risk of hospitalisation was lower in the case of two Omicron clades when compared to all other common clades which were detected in Estonia during the study period. There were no differences in hospitalisation rates between all the other clades. We can see slightly increased hospitalisation rates in relation to Omicron-infected young people (those under the age of twenty) when compared to other clades, with this being a trend which has been spotted by others [24]. Importantly, our data suggest that vaccination substantially reduced the probability of hospitalisation, especially in older populations which were at a higher risk of contracting severe levels of illness, irrespective of the involved virus clade. The vaccination effect on hospitalisation rates was strongly related to age, and tended to wane in younger age groups. We can speculate that the increasing protective effect of vaccination on hospitalisation rates in connection with age, particularly in those who were aged forty and above, can at least partially be attributed to a higher adaptive immune response to SARS-Cov-2 [26].

### The significance and implementation of SARS-CoV-2 sequencing

As in many countries, Estonia initiated SARS-CoV-2 sequencing in order to characterise the mutation patterns of the virus, and to complement the epidemiological data of outbreak management, including contact tracing, against viral genetic information. In 2021 the detection of new variants and potentially more virulent strains which were circulating around the world became the main focus of sequencing. This information was constantly integrated into decision-making by the Estonian Health Board and the government. Simultaneously, it is pivotal to acknowledge that, most of the time, we were sequencing the minimum level—or even higher—of samples as recommended by ECDC, enabling us to describe the variants almost in real-time in the background of the constantly and rapidly evolving COVID-19 pandemic.

We emphasise that, based upon an in-depth analysis of SARS-CoV-2 sequencing outcomes, it is possible to devise a sequencing strategy for future crises. For instance, the SARS-CoV-2 sequencing project made it possible for us to work out, from scratch, an effective sequencing approach at the national level, however, the downstream analysis further highlighted the significance of the proper stratification of samples in order to sufficiently cover various population groups and societal segments. Countries around the globe, including Estonia, expended substantial resources in conducting SARS-CoV-2 sequencing. Therefore, it is crucial to assess the optimal means by which such information can be acquired in the future. Our study provides valuable information for formulating areas of strategy in the event of a future pandemic which resembles the COVID-19 pandemic. Data which have been generated during this project could be integrated into modelling efforts which are aimed at enhancing levels of preparedness for future pandemics or epidemics.

### Strengths and limitations

The major strength of this study includes complementing our sequencing data with hospitalisation and vaccination data which has been retrieved from our national health databases, giving us a unique opportunity to explore the associations between viral genetics and disease outcomes. Our study has several limitations, including the uneven coverage of sample metadata, with less metadata being available for those samples which were collected at the beginning of the pandemic; and time-dependent confounding from multiple factors which can be related to different sets of public health mitigation measures which were being implemented during the consecutive virus waves, along with acquired immunity from SARS-CoV-2 infections, vaccine availability, and vaccination coverage, which together may have obscured the associations between clades and hospitalisation rates. Some of the analyses were affected by small sample sizes, such as when we pooled all the various vaccines in order to assess vaccine protection levels. Additionally, due to the small sample size, we could not reliably analyse the association of SARS-CoV-2 clades with the use of intensive care units or with mortality rates in Estonia.

### Conclusions

This national SARS-CoV-2 sequencing study from the first two years of the COVID-19 pandemic (March 2020 to March 2022) revealed the following:

1) disease waves in Estonia were driven by the Alpha, Delta, and Omicron clades.

2) those variants which belonged to the Omicron clade caused substantially less severe levels of illness and hospitalisation when compared to preceding variants.

3) the COVID-19 vaccination effect on hospitalisation rates was independent of the involved SARS-CoV-2 clade.

During the COVID-19 pandemic, such data helped state agencies to understand how SARS-CoV-2 spread around Estonia and how measures to tackle the pandemic could be implemented. The study also made it possible to follow the evolution of the virus during the pandemic. When the Omicron wave came into Estonia, the virus spread faster, but the disease had become significantly milder by then so strict restrictions to control the spread of the virus were no longer necessary.

## Supporting information

**S1 Table. Distribution of SARS-CoV-2 Pango lineages by the epidemiological wave.** Epidemiological waves were defined by the Estonian Health Board. Values are counts and percentages relative to the column are shown in parentheses.
(DOCX)

**S1 Fig. Age-group prevalence and dynamics of SARS-Cov-2 infections.** (A) the study population age group distribution relative to the 2022 general population. Points denote the best estimate of the aggregated binomial model, N = 26,618. Thick and thin lines denote 67% and 95% credible intervals, respectively. Dotted horizontal lines denote age group prevalence in the general population. (B) trends in the study population age group distribution during the study period. Lines denote the best estimate of the simple binomial model fitted to the weekly age group prevalence. Dashed lines denote unobserved dates implied by the underlying model. Ribbons denote a 95% credible interval. Solid horizontal lines denote SARS-Cov-2 waves. Panel A model summary is presented in Table 5 in S2 Appendix. Panel B model summary is presented in Table 6 in S2 Appendix.
(TIF)

**S2 Fig. Trends of sex balance among age groups.** Lines denote the best estimate of the Bernoulli model, N = 24,343. The dashed lines denote unobserved dates implied by the underlying model. Ribbons denote a 95% credible interval. Solid horizontal lines denote SARS-Cov-2 waves. The model summary is presented in Table 7 in S2 Appendix.
(TIF)

**S3 Fig. The proportion of domestic and travel-related cases.** (A) the weekly proportions of domestic- and travel-related cases and cases where such info was missing. (B) the association of travel-related cases with VOC indicated by the Bernoulli model, adjusted for the proportion of cases with missing info and population vaccination coverage, N = 14,341. (C) VOC association with countries or territories of travel-related cases indicated by the aggregated binomial model, N = 13,603. Y-axis labels are country ISO codes, and "Other" includes countries or territories associated with less than 30 cases. In panels B and C, points denote the model's best estimate, and thin and thick lines denote 66% and 95% credible intervals, respectively. Panel B model summary is presented in Table 8 in S2 Appendix. Panel C model summary is presented in Table 9 in S2 Appendix.
(TIF)

**S4 Fig. Association of average RT-PCR Cq values with SARS-CoV-2 clades.** Posterior summaries of the average Cq value of ORF1a, S gene, and N gene, obtained from a linear model adjusted for age, sex, and vaccination status, skew-normal distribution, N = 5,689. S gene dropouts were treated as missing data and only ORF1a and N gene Ct values were used for these samples. Point denotes the model's best fit and error bars denote a 95% credible interval. The model summary is presented in Table 10 in S2 Appendix.
(TIF)

**S5 Fig. Diversity of SARS-CoV-2 lineages in domestic and travel-related cases in Estonia.** (A) the observed number of Pango lineages. (B) the difference in the observed number of Pango lineages, imported cases compared to domestic cases. (C) evenness index. (D) the effect size of the evenness index in imported cases compared to domestic cases. In panels A and C, points denote individual weekly observations. The line denotes the autoregressive model best fit and the shaded ribbon denotes a 95% credible interval. In panels B and D, the line denotes effect size derived from the model fit shown in panels A or C, respectively, and the shaded ribbon denotes a 95% credible interval. Panel A and B model summary is presented in Table 11 in S2 Appendix. Panel C and D model summary is presented in Table 12 in S2 Appendix.
(TIF)

**S6 Fig. SARS-CoV-2 infected males show a higher probability of hospitalization independent of vaccination status.** (A) the probability of hospitalization is associated with sex, age, and vaccination status. (B) the odds ratio of hospitalization of males relative to females is associated with age, but independent of vaccination status. Posterior distributions of the estimated marginal means were obtained from the same model as in Fig 5. The line denotes the model's best estimate and the ribbon denotes a 95% credible interval. The dotted line denotes the log odds ratio = 0. The model summary is presented in Table 4 in S2 Appendix.
(TIF)

**S7 Fig. Lower hospitalization of Omicron-infected people among unvaccinated population.** The odds ratios of hospitalization among the unvaccinated population, pairwise comparisons of the SARS-CoV-2 clades, associated with age, averaged over both sexes. Posterior distributions of the estimated marginal means were obtained from the same model as in Fig 5. The line denotes the model's best estimate and the ribbon denotes a 95% credible interval. The dotted line denotes the log odds ratio = 0. The model summary is presented in Table 4 in S2 Appendix.
(TIF)

**S8 Fig. Lower hospitalization of Omicron-infected people among the vaccinated population.** The odds ratios of hospitalization among the vaccinated population, pairwise comparisons of the SARS-CoV-2 clades, associated with age, averaged over both sexes. Posterior distributions of the estimated marginal means were obtained from the same model as in Fig 5. The line denotes the model's best estimate and the ribbon denotes a 95% credible interval. The dotted line denotes the log odds ratio = 0. The model summary is presented in Table 4 in S2 Appendix.
(TIF)

**S1 Appendix. Supporting methods.**
(DOCX)

**S2 Appendix. Supporting tables.**
(XLSX)

## Author Contributions

**Conceptualization:** Aare Abroi, Radko Avi, Irja Lutsar, Kristi Huik.

**Data curation:** Heiki Niglas, Steven Smit, Erik Jaaniso, Ivan Kuzmin, Heleri Inno, Uku Raudvere.

**Formal analysis:** Taavi Päll.

**Funding acquisition:** Lili Milani, Mait Metspalu, Irja Lutsar.

**Investigation:** Taavi Päll, Arina Shablinskaja, Merit Pauskar, Ene-Ly Jõgeda, Hiie Soeorg, Eveli Kallas, Andrio Lahesaare, Kai Truusalu, Dagmar Hoidmets, Kaspar Ratnik, Hanna Sepp, Liidia Dotsenko, Jevgenia Epštein, Heleene Suija, Tuuli Reisberg.

**Methodology:** Radko Avi, Ulvi Gerst Talas.

**Project administration:** Katrin Kaarna, Kristi Huik.

**Resources:** Heiki Niglas, Andrio Lahesaare, Olga Sadikova, Kaspar Ratnik, Hanna Sepp, Liidia Dotsenko, Jevgenia Epštein, Heleene Suija, Katrin Kaarna, Lili Milani, Mait Metspalu, Mari-Anne Härma, Paul Naaber.

**Software:** Taavi Päll, Steven Smit, Ott Eric Oopkaup, Ivar Koppel, Ivan Kuzmin, Heleri Inno, Uku Raudvere, Hedi Peterson, Ulvi Gerst Talas.

**Supervision:** Ivar Koppel, Irja Lutsar, Kristi Huik.

**Visualization:** Taavi Päll.

**Writing – original draft:** Taavi Päll, Kristi Huik.

**Writing – review & editing:** Taavi Päll, Aare Abroi, Hiie Soeorg, Paul Naaber, Ulvi Gerst Talas, Irja Lutsar, Kristi Huik.

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
