## [Decision Letter · Decision Letter 0]

12 Feb 2024

PONE-D-23-26349SARS-CoV-2 clade dynamics and their associations with hospitalization during the first two years of the COVID-19 pandemicPLOS ONE

Dear Dr. Päll,

Thank you for submitting your manuscript to PLOS ONE. After careful consideration, we feel that it has merit but does not fully meet PLOS ONE’s publication criteria as it currently stands. Therefore, we invite you to submit a revised version of the manuscript that addresses the points raised during the review process.

We look forward to receiving your revised manuscript.

Kind regards,

Nihad A.M Al-Rashedi

Academic Editor

PLOS ONE

Journal Requirements:

Additional Editor Comments:

Your manuscript, "SARS-CoV-2 clade dynamics and their associations with hospitalization during the first two years of the COVID-19 pandemic" (PONE-D-23-26349), has been assessed by our reviewers. Although it is of interest, we are unable to consider it for publication in its current form. The reviewers have raised a number of points that we believe would improve the manuscript and may allow a revised version to be published in PLOS ONE.

Reviewers' comments:

Reviewer's Responses to Questions

**Comments to the Author**

1. Is the manuscript technically sound, and do the data support the conclusions?

Reviewer #1: Yes

Reviewer #2: Yes

2. Has the statistical analysis been performed appropriately and rigorously? 

Reviewer #1: Yes

Reviewer #2: Yes

3. Have the authors made all data underlying the findings in their manuscript fully available?

Reviewer #1: Yes

Reviewer #2: Yes

4. Is the manuscript presented in an intelligible fashion and written in standard English?

Reviewer #1: Yes

Reviewer #2: Yes

5. Review Comments to the Author

Reviewer #1: Päll et al. present a comprehensive study on the dynamics of SARS-CoV-2 variants during the first two years of the COVID-19 pandemic in Estonia, examining their correlation with hospitalization and exploring the potential effects of vaccination across different age and sex groups. The article is well-organized, maintaining a good flow and clarity despite the abundance of data. The methods, results, and conclusions are consistently presented. Several findings align with global publications, providing clear data and evidence that contribute to broader conclusions applicable to various geographic regions during the pandemic. However, I have a few comments:

In the summary, the statement "The odds of hospitalization in the vaccinated population decreased with age" might be prone to misinterpretation. The wording could be clearer and less confusing. It's essential to contextualize this statement. The conclusion in the summary appears better worded, emphasizing that the impact of vaccination might be more pronounced in older individuals, as evidenced by a decrease in hospitalization rates. In the results section, the wording slightly changes when compared to the unvaccinated population. However, there is room for improvement to prevent potential confusion.

Reviewer #2: The study by Päll et al. aims to identify the circulating SARS-CoV-2 variants and subsequently relate their diversity and hospitalization outcomes from the first two years of the COVID-19 pandemic in Estonia. The authors concluded that the Omicron variant is linked to a lower probability of hospitalization, and vaccination is protective against severe disease progression in people aged 40 years and older. Overall, the manuscript is well-written. However, there are several minor concerns that need to be addressed by the authors.

1. Please include justification for why this study is needed in the Introduction section.

2. Table 1 is confusing. Do the numbers in parentheses represent percentages or standard deviations? Please clarify.

3. Please consider separating the Discussion section into several paragraphs for better readability.

4. One of the interesting findings of this study is that the protective effect of vaccination against severe disease, associated with hospitalization, was observed from age 40 years onwards and this effect increased with age. Please elaborate more on this. Is there a possible explanation for this phenomenon?

5. Please include a paragraph on how the data in this study could be beneficial to the scientific community or government policymaking.

6. PLOS authors have the option to publish the peer review history of their article (what does this mean?). If published, this will include your full peer review and any attached files.

Reviewer #1: No

Reviewer #2: No

---

## [Author Response · Author response to Decision Letter 0]

8 Apr 2024

Dear Nihad A.M Al-Rashedi,

thank you and the reviewers for the comments and suggestions. Here we try to address the issues raised during the review. 

We updated title page formatting, referencing to figures and tables, and reference formatting according to PLOS One style.

2. We suggest you thoroughly copyedit your manuscript for language usage, spelling, and grammar. 

The manuscript was copyedited by a professional service:

Ref: Maarja Yörük

Tõlkebüroo Esteet OÜ

+372 56995633

www.esteet.ee

3. Please note that funding information should not appear in any section or other areas of your manuscript. We will only publish funding information present in the 

Funding Statement section of the online submission form. Please remove any funding-related text from the manuscript. 

All funding-related text was removed from the Manuscript.

We included the Ethics statement section under the Methods section and removed the ethics statement from other sections.

5. Please include captions for your Supporting Information files at the end of your manuscript, and update any in-text citations to match accordingly. Please see our Supporting Information guidelines for more information: http://journals.plos.org/plosone/s/supporting- information. 

We included S1 and S2 Appendix captions at the end of the Manuscript. Additionally, we restructured Supporting Information – moved the S1 Table to a separate file, moved the S2 Table to the S2 Appendix and updated in-text references accordingly. We moved the S1 Table, and S1-S8 Fig captions to the end of the Manuscript. 

Reviewer #1: Päll et al. present a comprehensive study on the dynamics of SARS-CoV-2 variants during the first two years of the COVID-19 pandemic in Estonia, examining their correlation with hospitalization and exploring the potential effects of vaccination across different age and sex groups. The article is well-organized, maintaining a good flow and clarity despite the abundance of data. The methods, results, and conclusions are consistently presented. Several findings align with global publications, providing clear data and evidence that contribute to broader conclusions applicable to various geographic regions during the pandemic. 

However, I have a few comments:

1) In the summary, the statement "The odds of hospitalization in the vaccinated population decreased with age" might be prone to misinterpretation. The wording could be clearer and less confusing. It's essential to contextualize this statement. The conclusion in the summary appears better worded, emphasizing that the impact of vaccination might be more pronounced in older individuals, as evidenced by a decrease in hospitalization rates. In the results section, the wording slightly changes when compared to the unvaccinated population. However, there is room for improvement to prevent potential confusion.

Our main goal in this sequencing study, regarding vaccination, was to analyse if vaccination benefit is related to virus clade, for which we could not find support. Whereas hospitalisation was strongly related to age and Omicron caused fewer hospitalisations. Vaccination benefits regarding hospitalisation were not the same in all age groups, instead in the over-forties population, where hospitalisation became considerable, vaccination benefit was reduced with younger age, correlating with findings by others that adaptive immune response to SARS-CoV-2 increases with age. To make these results clearer, we updated the text, respectively, in the Abstract:

“Hospitalisation increased markedly with age in the over-forties, and was negligible in the under-forties. Vaccination decreased the odds of hospitalisation in over-forties. The effect of vaccination on hospitalisation rates was strongly dependent upon age but was clade-independent.”

in the Results section: 

“We found that the probability of hospitalisation was strongly related to age and COVID-19 vaccination status (Fig 5A). Overall, the probability of hospitalisation was very low for people who were under forty years of age but started to increase after that, whereas vaccinated people had a lower probability of being hospitalised when compared to unvaccinated people. The beneficial effect of any vaccination increased considerably with age, as the odds of hospitalisation decreased with age in the vaccinated population when compared to the unvaccinated (OR = 0.71, 95% CI [0.58, 0.84] at forty years of age; OR = 0.37, 95% CI [0.3, 0.45] at sixty, and OR = 0.19, 95% CI [0.13, 0.27] at eighty. This association between age and vaccination effect was similar across all common clades (Fig 5B).”

and the Discussion:

“Importantly, our data suggest that vaccination substantially reduced the probability of hospitalisation, especially in older populations which were at a higher risk of contracting severe levels of illness, irrespective of the involved virus clade. The vaccination effect on hospitalisation rates was strongly related to age, and tended to wane in younger age groups. We can speculate that the increasing protective effect of vaccination on hospitalisation rates in connection with age, particularly in those who were aged forty and above, can at least partially be attributed to a higher adaptive immune response to SARS-Cov-2 [26].”

Reviewer #2: The study by Päll et al. aims to identify the circulating SARS-CoV-2 variants and subsequently relate their diversity and hospitalization outcomes from the first two years of the COVID-19 pandemic in Estonia. The authors concluded that the Omicron variant is linked to a lower probability of hospitalization, and vaccination is protective against severe disease progression in people aged 40 years and older. Overall, the manuscript is well-written. However, there are several minor concerns that need to be addressed by the authors.

1. Please include justification for why this study is needed in the Introduction section.

We rewrote the end of the Introduction to better address the justification of this study: “We undertook this study in order to be able to understand how distinct SARS-CoV-2 clades influenced the severity of COVID-19 and hospitalisation, and which other factors - including immunisation against SARS-CoV-2 - served to contribute to this process.”

2. Table 1 is confusing. Do the numbers in parentheses represent percentages or standard deviations? Please clarify.

The dimension of the value in parentheses is indicated in the first column of Table 1, it’s either an SD, % or IQR, depending on the type of the data. We updated the Table 1 legend to improve table readability.

3. Please consider separating the Discussion section into several paragraphs for better readability.

Thank you for the suggestion, we split the Discussion into separate paragraphs.

4. One of the interesting findings of this study is that the protective effect of vaccination against severe disease, associated with hospitalization, was observed from age 40 years onwards and this effect increased with age. Please elaborate more on this. Is there a possible explanation for this phenomenon?

We don’t have a clear explanation for this, nor does our data allow us to address this question directly. However, there seems to be a higher adaptive immune response to SARS-Cov-2 in older individuals (doi:10.1126/scitranslmed.abd5487), which seems to be a good working hypothesis. We amended the Discussion accordingly: 

“We can speculate, that the increasing protective effect of vaccination on hospitalization with age, particularly in those over 40 years old, can be at least partly attributed to higher adaptive immune response to SARS-Cov-2 [26].” 

5. Please include a paragraph on how the data in this study could be beneficial to the scientific community or government policymaking.

We amended the Discussion with the ‘Significance and implementation of SARS-CoV-2 sequencing section’, to better address aspects of the study relevant to government policymaking: 

“As in many countries, in Estonia, we initiated SARS-CoV-2 sequencing to characterize the mutation patterns of the virus, and to complement the epidemiological data of outbreak management, including contact tracing, with viral genetic information. In 2021, the detection of new variants and potentially more virulent strains circulating in the world became the main focus of the sequencing. This information was constantly integrated into the decision-making by the Estonian Health Board and the Government. Simultaneously, it is pivotal to acknowledge that most of the time we sequenced the minimum or even higher proportion of samples recommended by ECDC, enabling us to describe the variants almost in real-time in the background of the constantly and rapidly evolving COVID-19 pandemic. In addition, as shown here, we assessed the viral diversity and the relationships of SARS-CoV-2 clades with hospitalization and vaccination.

We emphasize that based on an in-depth analysis of SARS-CoV-2 sequencing outcomes, it is possible to devise a sequencing strategy for future crises. For instance, the SARS-CoV-2 sequencing project enabled us to work out from scratch an effective sequencing approach at the national level, and downstream analysis brought out the significance of proper stratification of samples to sufficiently cover various population groups and societal segments, that should be prioritized for sequencing. Countries around the globe, including Estonia, expended substantial resources in conducting SARS-CoV-2 sequencing; hence, it is crucial to assess the optimal means by which such information can be acquired in the future. Our study provides valuable information for formulating strategies in the event of a future pandemic resembling COVID-19. The data generated during this project could be integrated into modelling efforts aimed at enhancing preparedness for future pandemics or epidemics.

”

Yours sincerely,

Taavi Päll

Research fellow of medical virology

Department of Microbiology, 

Institute of Biomedicine and Translational Medicine, 

University of Tartu, 

Ravila 19

50411 Tartu, 

Estonia

Email: taavi.pall@ut.ee

Mob.: +372 56452732

---

## [Decision Letter · Decision Letter 1]

22 Apr 2024

SARS-CoV-2 clade dynamics and their associations with hospitalization during the first two years of the COVID-19 pandemic

PONE-D-23-26349R1

Dear Dr. Päll,

We’re pleased to inform you that your manuscript has been judged scientifically suitable for publication and will be formally accepted for publication once it meets all outstanding technical requirements.

Kind regards,

Nihad A.M Al-Rashedi

Academic Editor

PLOS ONE

Additional Editor Comments (optional):

Reviewers' comments:

Reviewer's Responses to Questions

**Comments to the Author**

1. If the authors have adequately addressed your comments raised in a previous round of review and you feel that this manuscript is now acceptable for publication, you may indicate that here to bypass the “Comments to the Author” section, enter your conflict of interest statement in the “Confidential to Editor” section, and submit your "Accept" recommendation.

Reviewer #1: All comments have been addressed

Reviewer #2: All comments have been addressed

2. Is the manuscript technically sound, and do the data support the conclusions?

Reviewer #1: Yes

Reviewer #2: Yes

3. Has the statistical analysis been performed appropriately and rigorously? 

Reviewer #1: Yes

Reviewer #2: Yes

4. Have the authors made all data underlying the findings in their manuscript fully available?

Reviewer #1: Yes

Reviewer #2: Yes

5. Is the manuscript presented in an intelligible fashion and written in standard English?

Reviewer #1: Yes

Reviewer #2: Yes

6. Review Comments to the Author

Reviewer #1: I consider the authors have made revisions that enhance the quality of the writing, making it suitable for publication

Reviewer #2: The authors have revised the manuscript accordingly. Additionally, the manuscript is now more readable after grammar editing. I suggest accepting it for publication in PLOS ONE.

7. PLOS authors have the option to publish the peer review history of their article (what does this mean?). If published, this will include your full peer review and any attached files.

Reviewer #1: No

Reviewer #2: No

---

## [Editor Report · Acceptance letter]

29 Apr 2024

PONE-D-23-26349R1 

PLOS ONE

Dear Dr. Päll, 

I'm pleased to inform you that your manuscript has been deemed suitable for publication in PLOS ONE. Congratulations! Your manuscript is now being handed over to our production team.

Kind regards, 

on behalf of

Dr. Nihad A.M Al-Rashedi 

Academic Editor

PLOS ONE